# Role of Exosomal miR-205-5p Cargo in Angiogenesis and Cell Migration

**DOI:** 10.3390/ijms25020934

**Published:** 2024-01-11

**Authors:** Miriam Martínez-Santos, María Ybarra, María Oltra, María Muriach, Francisco J. Romero, Maria E. Pires, Javier Sancho-Pelluz, Jorge M. Barcia

**Affiliations:** 1Escuela de Doctorado, Universidad Católica de Valencia San Vicente Mártir, 46001 Valencia, Spain; miriammartinez@mail.ucv.es (M.M.-S.); maria.ybarra@ucv.es (M.Y.); me.dossantos@ucv.es (M.E.P.); jm.barcia@ucv.es (J.M.B.); 2Centro de Investigación Translacional San Alberto Magno, Universidad Católica de Valencia San Vicente Mártir, 46001 Valencia, Spain; maria.oltra@ucv.es; 3Facultad de Medicina y Ciencias de la Salud, Universidad Católica de Valencia San Vicente Mártir, 46001 Valencia, Spain; 4Facultad de Ciencias de la Salud, Universidad Jaime I, Avda. Vicent Sos Baynat, 12006 Castellón de la Plana, Spain; muriach@med.uji.es; 5Hospital General de Requena, Conselleria de Sanitat, Generalitat Valenciana, 46340 Requena, Spain; jromerogomez@gmail.com

**Keywords:** extracellular vesicles, miR-205-5p, angiogenesis, retinal pigment epithelium, migration

## Abstract

Exosomes or small extracellular vesicles (sEVs) represent a pivotal component in intercellular communication, carrying a diverse array of biomolecules. Several factors can affect sEVs release dynamics, as occurs in hyperglycemia or inflammation. In fact, sEVs release has been associated with the promotion of physio-pathological processes. Among the sEVs cargo, microRNAs play an essential role in cell-to-cell regulation. More concretely, miR-205-5p is related to angiogenesis and cell proliferation. The aim of this study is to understand the specific role of sEVs containing miR-205-5p under high glucose conditions. ARPE-19 cells were cultured with high glucose (HG) for 5 days. sEVs were isolated and characterized. sEVs from ARPE-19 were used for angiogenesis and cell proliferation. HG increased sEVs release but downregulated miR-205-5p cargo expression compared to the control. sEVs from HG-treated ARPE-19 cells promoted tube formation and migration processes. In contrast, miR-205-5p overexpression (by mimic transfection) decreased angiogenesis and cell migration. Our results demonstrate how ARPE-19 cells respond to HG challenge by increasing sEVs with weak miR-205-5p cargo. The absence of this miRNA in sEVs is enough to promote angiogenesis. In contrast, restoring sEVs-miR-205-5p levels decreased it. These findings open new possibilities in sEVs-based therapies containing miR-205-5p against angiogenesis.

## 1. Introduction

Exosomes or small extracellular vesicles (sEVs) are spherical lipidic membranes (30–150 nm diameter) released by cells containing biomolecules as proteins, lipids, and nucleic acids. sEVs represent a pivotal component in cell to cell communication [1]. Among their nucleic acid content, microRNAs (miRNAs) are notably significant, constituting approximately 40% of the total sEVs-RNA cargo [2]. Once internalized through endocytosis by recipient cells, miRNAs play a crucial role in modulating cellular functions [3]. Generally, miRNAs regulate gene transcription, modulating key biological pathways, e.g., angiogenesis or cell migration [4].

Under oxidative stress conditions, cells undergo significant physiological alterations, increasing sEVs release [5]. These sEVs may present altered miRNAs cargo, probably due to the inability to encapsulate them in a physiological manner [6]. This phenomenon has been observed in some diseases as diabetes [7], where oxidative stress plays a crucial role in sEVs release, composition and function [8].

Epithelial cells typically release sEVs as part of their physiology. In fact, retinal pigment epithelium (RPE) regulates choroidal endothelial cells through sEVs and other direct interactions in retina [9]. These vesicles can induce significant changes in endothelial cell morphology and function, affecting retinal vascular dynamics [10]. Recent studies suggest that sEVs derived from ARPE-19 (a human RPE cell linage) contain a variety of bioactive factors capable of modifying vascular permeability [11].

Among miRNAs included as sEVs cargo, miR-205-5p has emerged as a key regulator in several processes such as angiogenesis [12]. Under a high glucose (HG) context, several miRNAs are dysregulated, including miR-205-5p. This fact has been related to vascular alterations and potentially to diabetic-related conditions such as diabetic retinopathy or nephropathy [13].

The therapeutic potential of non-coding RNAs against vascular proliferation and cell migration is currently being explored in cancer and other diseases [14]. The aim of this study is to understand the role of miR-205-5p as sEVs cargo under HG conditions in angiogenesis and cell migration.

## 2. Results

### 2.1. sEVs Isolation and Characterization 

sEVs size distribution was analyzed by a nanoparticle tracking system (Nanosight). Nanotracking revealed that most of sEVs ranged between 100 and 180 nm, with a maximum peak around 100 nm. TEM images confirmed the presence of spherical cup-like shaped sEVs with an approximate diameter of 100 nm (Figure 1A,B). After 35 mM glucose treatment for 5 days, ARPE-19 cells released a significantly higher number of sEVs (3.74 × 10^8^ ± 1.30 × 10^7^ particles/mL) compared to the control (1.37 × 10^8^ ± 6.28 × 10^7^ particles/mL) (Figure 1C). The presence of sEVs-specific proteins (CD9 and CD63) further confirmed the identification of sEVs. These two protein markers were positive for sEVs in control and in 35 mM glucose cell culture media (Figure 1D). As the negative control, β -actin and cell culture medium supernatant were used. The lack of β-actin in the cell supernatant compared to cell lysate indicated good separation.

### 2.2. sEVs from ARPE-19 Are Incorporated into HUVEC 

One of the first aspects was aimed to resolve whether sEVs are incorporated into HUVEC. For this, vesicle uptake assay based on PKH26 labeling was conducted on HUVEC. sEVs released from ARPE-19 were labeled with PKH26 and added to the HUVEC cell culture. Fitting with sEVs number release, HUVEC treated with control sEVs (Figure 2A) exhibited lower uptake compared to sEVs from 35 mM glucose conditions (Figure 2B). sEVs’ uptake was time-dependent, being almost absent at time cero, weak at 8h and maximum at 12 h (Figure 2C,F). A higher resolution with confocal microscopy was conducted to observe PKH26^+^/sEVs deposits. Sparse spot-like labeling could be found with perinuclear and cytoplasmic location (Figure 2D,E).

### 2.3. miR-205-5p Cargo in sEVs

miR-205-5p has been related to angiogenesis and can be included in sEVs. The condition 35mM glucose promoted higher sEVs release compared to the control. Surprisingly, HG-sEVs presented lower miR-205-5p compared to control sEVs (0.32 ± 0.19) (Figure 3A). As miR-205-5p was significantly decreased in 35 mM glucose sEVs, the next experiment was aimed to increase (artificially) miR-205-5p levels inside sEVs. ARPE-19 cells were transfected with miR-205-5p mimic after 35 mM glucose exposure. As expected, the addition of the miR-205-5p mimic significantly increased miR-205-5p levels in both, control (1654 ± 828) and 35 mM glucose conditions (1183 ± 553) (Figure 3B), thus confirming the incorporation of miR-205-5p into sEVs.

### 2.4. miR-205-5p sEVs Cargo Level Is Inversely Related to Angiogenesis and Migration

HG can promote angiogenesis by inducing the expression of pro-angiogenic factors. In order to study the role of sEVs containing miR-205-5p in angiogenesis, matrigel assay was performed. HUVEC treated with sEVs from ARPE-19 cell culture medium were used. HUVEC cultured with control sEVs presented few tube-like structures connecting neighboring cells, measured as total length %. Little length connection could be observed under control sEVs (Figure 4A) with large unfilled lagoons. However, HUVEC tube formation (Figure 4C), including all of the parameters: total length (Figure 4E), mesh index (Figure 4F), number of master junctions (Figure 4G) and segments (Figure 4H) was significantly increased by sEVs obtained from glucose treated cells (139.90 ± 7.09) (Figure 4C). This effect due to HG exposure was significantly attenuated by miR-205-5p mimic transfection: control + mimic (miR-205-5p) (96.45 ± 1.642) (Figure 4B) and 35 mM glucose + mimic (miR-205-5p) (97.41 ± 6.70). (Figure 4D). All parameters related to tube formation, total length, mesh index, number of master junctions and segments were also restored to control values in 35 mM conditions (Figure 4E,H). There is a lack of difference when comparing control vs control + mimic. These results indicate that miR-205-5p restores tube formation under high glucose conditions and not under control ones.

Cell migration was also analyzed by wound healing assay under the same sEVs conditions in ARPE-19. sEVs from 35 mM glucose (49.33 ± 15.88) resulted in faster wound closure compared to control sEVs (31.75 ± 13.88) (Figure 5A,C). This effect was normalized by ectopic addition of the miR-205-5p mimic, resulting in a lower migration rate (Figure 5B,D) and attenuating the percentage of the migrated area and gap distance (Figure 5E,F). Fitting with those data from tube formation, there is no difference in cell migration when comparing control vs. control + mimic, supporting the idea of the specific role of miR-205-5p under high glucose conditions.

## 3. Discussion

sEVs release dynamics are significantly influenced by the cellular microenvironment. sEVs cargo and composition change with environmental stimuli [9]. Recent studies have demonstrated that under oxidative stress conditions such as hyperglycemia, cells tend to release a greater number of sEVs [15]. sEVs release is considered an adaptive cellular communication mechanism, allowing cells to respond and adapt to environmental changes. Particularly, hypoxia and hyperglycemia not only increase sEVs release but also modify its content, affecting neighboring cells [16,17]. This phenomenon has been associated with the promotion of critical physiological and pathological processes such as autophagy, cell migration, angiogenesis, and endothelial dysfunction [18,19,20]. Fitting with this, we demonstrate how HG exposure significantly increases sEVs release from ARPE-19 cells similarly to other oxidative challenges [9,15,16,21,22,23]. 

The mechanisms behind how sEVs dock to recipient cells is not fully understood and it depends on cell type and other factors [24]. Thus, sEVs can be fused directly to recipient cells or they can be engulfed by endocytosis [25,26,27,28]. Even more, cell recipient type results determinant for sEVs incorporation, different reports indicate cell-target preference on this sEVs docking [26,29,30]. According to our results, recipient HUVEC completely include sEVs released from ARPE-19 cells. Confocal images indicate apparent perinuclear endoplasmic-like location of sEVs, fitting with other reports [16,18,19]. This fact suggests endocytosis as part of the sEVs docking mechanism. Future studies will be conducted to understand in detail the precise sEVs location and final destination in recipient cells. 

Previous data from our laboratory (under review) indicated that 35 mM glucose-conditioned ARPE-19 cell culture medium increased tube formation in HUVEC. More specifically, the results herein indicate that just sEVs are enough to modulate this phenomenon in endothelial cells (Figure 4 and Figure 5). It is noteworthy that miR-205-sEVs cargo is essential for tube formation/inhibition and cell migration under high glucose conditions. Interestingly, there is a lack of miR-205-5p effect under control conditions (control vs. control + mimic). This discrepancy can be explained since angiogenesis is promoted under physio-pathological conditions, e.g., high glucose. This scenario activates oxidative stress-related pathways, such as HIF-1α/VEGFA, also decreasing miR-205-5p levels. As miR-205-5p targets the HIF-1α/VEGFA axis, cell migration/proliferation is promoted under oxidative conditions. However, under control conditions, HIF-1α/VEGFA is not overexpressed and thus the addition of miR-205-5p cannot be appreciated in terms of cell migration/proliferation. 

In spite of the fact that HG promoted higher sEVs release, they exhibited lower miR-205-5p levels compared to control sEVs. Isolated sEVs with low miR-205-5p cargo were capable of promoting angiogenic changes. In contrast, sEVs with higher miR-205-5p content inhibited tube formation. (Figure 4 and Figure 5). One step further, we demonstrate the effect of miR-205-5p by increasing miR-205-5p levels (mimic transfection). As expected after transfection, we have found a significant increase in sEVs containing miR-205-5p in all conditions. This indicates that ectopic miR-205-5p is packaged into sEVs and also confirms that sEVs containing high miR-205-5p levels inhibit tube formation. In contrast, sEVs containing lower miR-205-5p levels significantly increase tube formation under HG conditions (Figure 4 and Figure 5). 

Consistent with our findings, previous studies have reported that sEVs with high miR-205-5p cargo inhibit angiogenesis by targeting one of the key elements for angiogenesis: vascular endothelial growth factor (VEGFA). Fitting with this, the results from our laboratory (under review) indicate that miR-205-5p decreased VEGFA and hypoxia inducible factor (HIF-1α) mRNA levels. Different studies have attributed anti-inflammatory properties to sEVs containing miR-205-5p in rheumatoid arthritis and chronic periodontitis [31]. However, in the cancer context, there are some discrepancies: sEVs containing miR-205-5p released in nasopharyngeal carcinoma and lung adenocarcinoma have demonstrated pro-angiogenic properties [32]. Meanwhile, miR-205-5p has been related to anti-angiogenesis in other cancer types [33] and even proposed as biomarker [34]. Plausibly, this controversy can be understood according to recipient specificity cell docking, apart from other cargo biomolecular content. 

As a conclusion, the presence of miR-205-5p within these sEVs can inhibit angiogenesis and migration as depicted in our proposed model (Figure 6). The specific role of miR-205-5p results enough eloquent to consider it as an alternative for VEGFA-based antibody treatment against vascular proliferative disorders.

## 4. Materials and Methods

### 4.1. Cell Culture

The human ARPE-19 cell line was procured from the American Type Culture Collection (ATCC) based in Manassas, VA, USA. These cells were maintained in Dulbecco’s Modified Eagle Medium/Nutrient Mixture F-12 (DMEM/F12) provided by Invitrogen. We utilized cells from the 11th to the 30th passages. The cells were grown until they reached 80–90% confluence, starting from a density of 1 × 10^6^ cells cm^2^ in various plates. ARPE-19 cells were transferred to either 24-well or 6-well plates and treated for 120 h. In the high glucose (HG) experimental group, the ARPE-19 cells were incubated with 35 mmol/L of glucose (sourced from Sigma-Aldrich, St. Louis, MO, USA). The control group (CG) was maintained in 5.5 mmol/L of glucose (Sigma-Aldrich, St. Louis, MO, USA), with 19.5 mmol/L of mannitol (Sigma-Aldrich, St. Louis, MO, USA). Human umbilical vein endothelial cells (HUVEC) were extracted from umbilical veins using established methodologies. These HUVEC were then cultured in endothelial cell media (sourced from Innoprot, Derio, Spain). Cells were maintained at 37 °C in a 5% CO_2_ environment.

### 4.2. sEVs Isolation

sEVs from ARPE-19 cell medium were isolated using the total exosome isolation reagent from the cell culture medium (Thermo Fisher Scientific, Waltham, MA, USA), following the previously described protocol for matrigel and migration assay [35]. For immunofluorescence imaging and electron microscopy, sEVs were isolated through a series of centrifugation steps. First, centrifugation at 700× *g* for 30 min was performed to remove cellular debris, followed by centrifugation at 14,000× *g* for 30 min to eliminate apoptotic bodies. Microvesicles were removed through centrifugation at 40,000× *g*. The final ultracentrifugation step at 150,000× *g* for 90 min was performed to obtain the sEVs.

### 4.3. sEVs Characterization

To confirm the identity of sEVs, the number of particles was quantified using the nanoparticle tracking system Nanosight NS300 (Malvern Instruments, Malvern, UK). Additionally, transmission electron microscopy (TEM) was employed to observe the morphology and size of the sEVs. Pellets containing sEVs from ARPE-19 cell culture media were fixed in 2.5% glutaraldehyde (pH 7.4) for 40 min, followed by post-fixation in 1% osmium tetroxide (pH 7.4) for 40 min. Subsequently, the samples were dehydrated using a series of ethanol concentrations. After dehydration, the samples were embedded in epoxy resin mixture, polymerized, sectioned into 70 nm slices, and observed under an electron microscope (FEI Tecnai G2 Spirit TWIN 120 kV, Fei Europe, Eindhoven, The Netherlands). 

### 4.4. Western Blot 

sEVs, treated or control, were resuspended in RIPA-50 buffer (Thermo Fisher Scientific, Waltham, MA, USA) and protease/phosphatase inhibitor cocktail (Sigma-Aldrich, St. Louis, MO, USA). Equal amounts of protein (70 µg) were loaded and measured by SDS-PAGE on 4–12% SDS-Polyacrylamide gel electrophoresis and electro-blotted onto a polyvinylidene difluoride membrane (PVDF; Millipore, Burlington, MA, USA) through wet transfer. After blocking with 5% skim milk, membranes were incubated overnight with CD9 (1:200; Santa Cruz Biotechnology, Santa Cruz, CA, USA) and CD63 antibodies (1:200; Santa Cruz Biotechnology, Santa Cruz, CA, USA). Finally, membranes were incubated for 1 h at room temperature with anti-mouse and anti-rabbit IgG-HRP antibodies (1:500 Santa Cruz Biotechnology, Santa Cruz, CA, USA). The visualization was performed with ECL (Pierce, Thermo Fisher Scientific, Waltham, MA, USA) and detected by Image Quant LAS-100 mini (GE Healthcare, Chicago, IL, USA). 

### 4.5. Fluorescence Labeling of sEVs and Internalization

ARPE-19 cell culture medium was collected for sEVs isolation. sEVs were obtained by differential ultracentrifugation. To label sEVs, 1 mL of Diluent C and 4 mL of PKH26 (Red Fluorescent Cell linker for General Cell Membrane, Sigma-Aldrich, St. Louis, MO, USA) were added to the sEVs pellet, followed by 1 h of incubation. The staining process was stopped by adding an equal volume of DPBS. Labeled sEVs were then subjected to ultracentrifugation at 120,000× *g* for 1 h and resuspended in exo-free serum cell media. HUVEC were cultured on a 24 mm circular coverslip up to 80% confluence, then mounted into the recording chamber. sEVs labeled with PKH26 from ARPE-19 medium were added to HUVEC. Pictures were acquired using a Leica TCS SP8 Inverted Confocal Microscope (Leica, Wetzlard, Germany) with 63X oil immersion objective and a Leica DM IL LED inverted microscope. The settings for detecting the PKH26 emission fluorescence were excitation at 550 nm and emission at 667 nm. Fluorescence quantification was performed using grayscale with the LasX software Office 1.4.5.

### 4.6. Mimic Transfection

ARPE-19 cells at 60–80% confluence were treated for 5 days and then transfected with miR-205-5p mirVana^®^ miRNA mimic (Thermo Fisher Scientific, Waltham, MA, USA) at a concentration of 30 pmol (10 µM) using Lipofectamine 2000 RNAiMAX Reagent (Thermo Fisher Scientific, Waltham, MA, USA) diluted in Opti-MEM^®^ Medium (Thermo Fisher Scientific, Waltham, MA, USA). After 48 h of transfection, the conditioned medium was collected for sEVs isolation and stored at −80 °C for further assays.

### 4.7. miRNA Expression Analysis

To assess the expression profiles of selected miRNAs in sEVs, quantitative real-time PCR (qRT-PCR) was employed. For the analysis of miRNA expression, RNA quantities ranging from 10 to 20 ng were reverse transcribed using the TaqMan MicroRNA Reverse Transcription Kit provided by (Applied Biosystems, Waltham, MA, USA. This process utilized specific TaqMan RT primers and was conducted in the Verity pro 96-well thermal cycler from (Applied Biosystems, Waltham, MA, USA) following a thermal cycling protocol of 16 °C for 30 min, 42 °C for 30 min, 85 °C for 5 min, and a hold at 4 °C. The qRT-PCR was carried out using TaqMan™ microRNA Assays (Thermo Fisher Scientific, Waltham, MA, USA) in conjunction with the TaqMan Gene Expression Master Mix (Applied Biosystems, Waltham, MA, USA). The RT-PCR was performed on a Roche 234 LighterCycler 480, adhering to specified temperature cycling conditions. For normalization purposes, RNU6B and RNU43 snoRNAs were used. The relative expression levels were determined using the 2^−ΔΔCt^ method.

### 4.8. Vasculogenesis Assay

Pre-cooled 96-well plates were coated with 70 μL of Matrigel (Becton Dickinson, Billerica, MA, USA). Human umbilical vein endothelial cells (HUVEC) were then plated at a concentration of 3.0 × 10^4^ cells per square centimeter and subsequently exposed to sEVs from ARPE-19 cells for a duration of 5 h. We added 70 μL of sEVs to each well. Matrigel was dropped for a 30 min period to solidify at 37 °C. For imaging, an Olympus CKX41 inverted microscope (Olympus, Tokyo, Japan) coupled with an Olympus DP74 digital camera was used. Total tube length was measured by ImageJ software 1.8.0 (National Institute of Health, Bethesda, MD, USA), employing the Angiogenesis Analyzer plugin for quantification.

### 4.9. Scratch Wound Healing Assay

ARPE-19 cells were seeded at a density of 5.0 × 10^4^ cells/cm^2^ in a 24-well plate and incubated at 37 °C and 5% CO_2_ for 48 h. A wound was created by manually scratching the cell monolayers using a sterile 200 µL pipette tip across the center of the well when the cells reached 90% confluence. Images were captured at 0, 4, 8 and 24 h using an Olympus CKX41 inverted microscope and recorded with an Olympus DP74 digital camera. The gap distance was analyzed using Image J software 1.8.0, and the total migrated area was measured in µm.

## Figures and Tables

**Figure 1 ijms-25-00934-f001:**
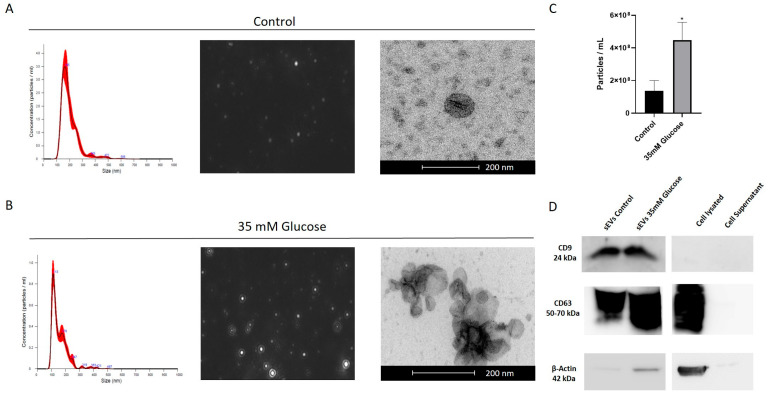
Characterization of sEVs released by ARPE-19 cells. Nanoparticle tracking analysis and TEM were used to analyze the size distribution, morphology and to quantify sEVs in ARPE-19 cells under both control conditions (**A**) and high glucose treatment (**B**). Scale bar 200 nm. Difference between number of control sEVs and 35 mM glucose sEVs is presented in particles/mL (**C**). Western blot analysis shows CD9, CD63 and β-Actin expression in sEVs, lysate and cell supernatant (**D**). Values are presented as mean ± SEM (*n* = 5). *p*-value was determined using *two-tailed t-test * p* < 0.05.

**Figure 2 ijms-25-00934-f002:**
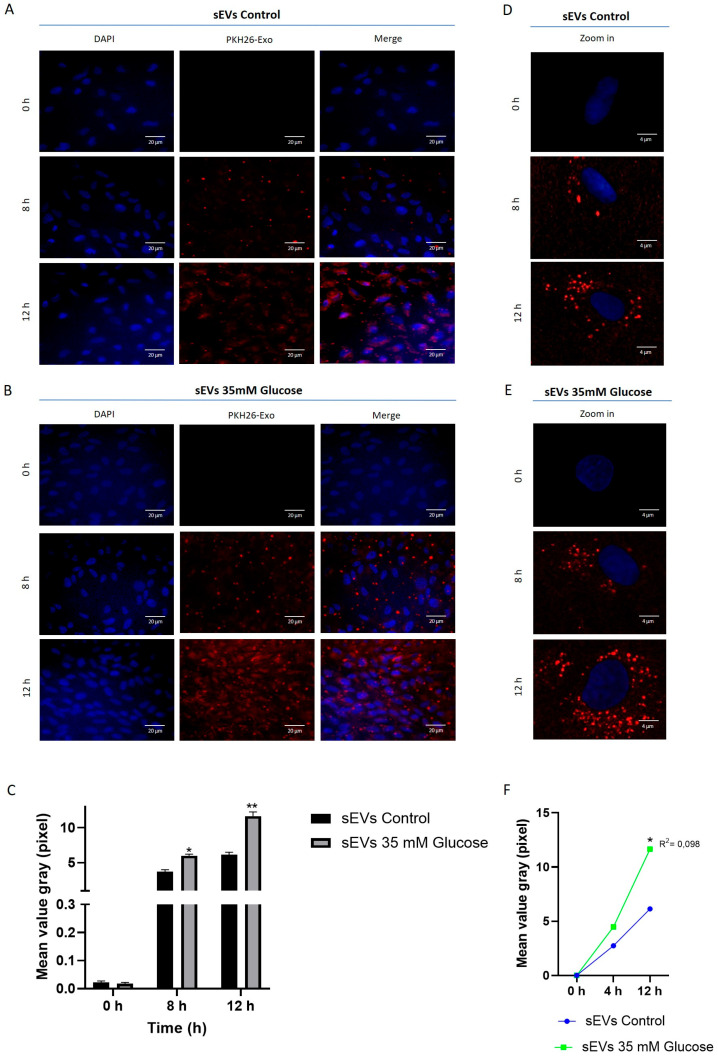
sEVs released by ARPE-19 cells are incorporated into HUVEC. sEVs released from ARPE-19 under control (**A**) or 35 mM glucose (**B**) were labeled with PKH26 (red) and then incubated with HUVEC. Nuclei stained with DAPI (blue). Representative images were taken at 0 h, 8 h and 12 h by using Leica DM IL LED inverted microscope. Scale bar: 100 nm. Quantification of fluorescence intensity was calculated using Leica Application Suite X (**C**). Confocal higher resolution images were acquired using a Leica TCS SP8 (**D**,**E**). The Spearman correlation shows the relationship between time and the amount of fluorescence emitted (**F**). Values are expressed as mean ± SEM (*n* = 3). *p*-value was obtained by *t*-test; * *p* < 0.05 and ** *p* < 0.01.

**Figure 3 ijms-25-00934-f003:**
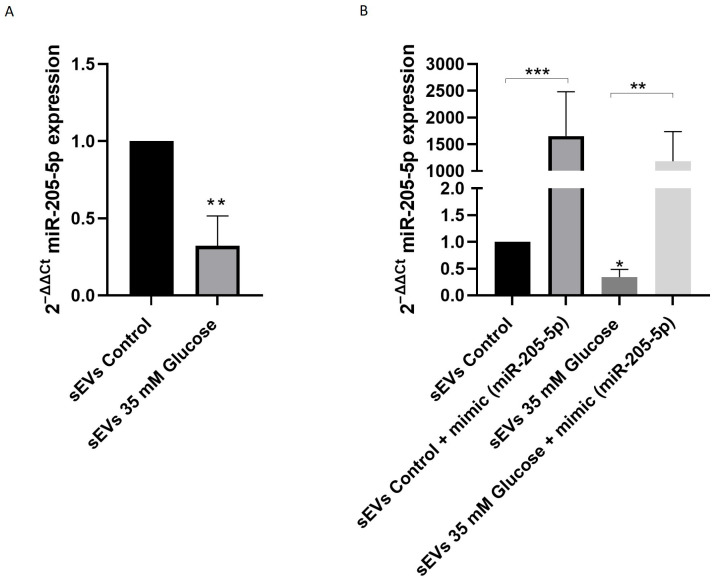
miR-205-5p is included in sEVs. miR-205-5p was detected in sEVs from 35 mM glucose and control ARPE-19 cells (**A**). Ectopic administration of miR-205-5p mimic significantly increased miR-205-5p in all transfected conditions (**B**). Values are presented as mean ± SEM (*n* = 4 and *n* = 4). *p*-value was determined using *t*-test and one-way ANOVA followed by the Kruskal–Wallis test, respectively; * *p* < 0.05, ** *p* < 0.01 and *** *p* < 0.001.

**Figure 4 ijms-25-00934-f004:**
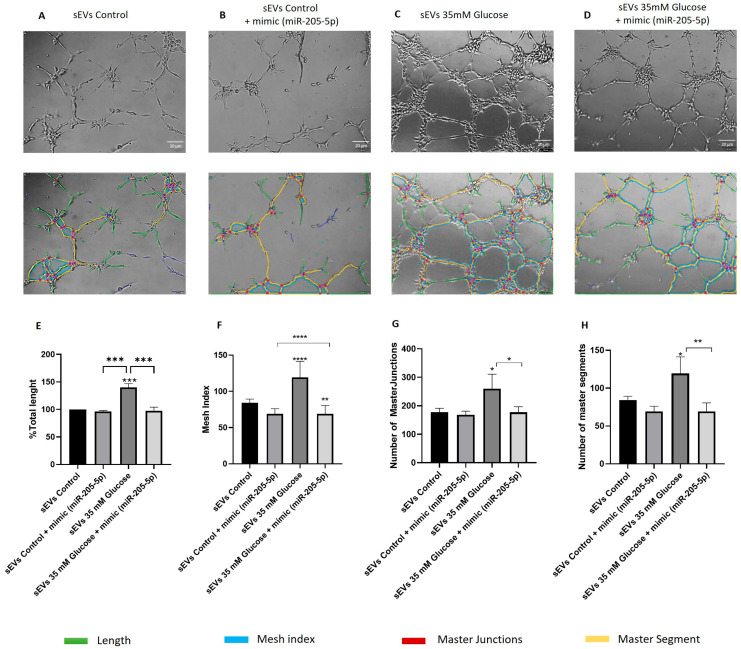
Tube formation was regulated by miR-205-5p in HUVEC cells. HUVEC tube formation was subject to isolated sEVs from different ARPE-19 sEVs cell culture media conditions. Control medium sEVs (**A**), control medium with mimic miR-205-5p sEVs (**B**), 35 mM glucose medium sEVs (**C**) and 35 mM glucose medium with mimic miR-205-5p sEVs (**D**). Total length (**E**), Mesh index (**F**), Number of masters junctions (**G**) and segments (**H**) were measured by Image J (version 1.8.0) Values are expressed as mean ± SEM (*n* = 3). *p*-value was obtained by one-way ANOVA followed by Tukey’s test: * *p* < 0.05, ** *p* < 0.01 and *** *p* < 0.001, **** *p* < 0.0001.

**Figure 5 ijms-25-00934-f005:**
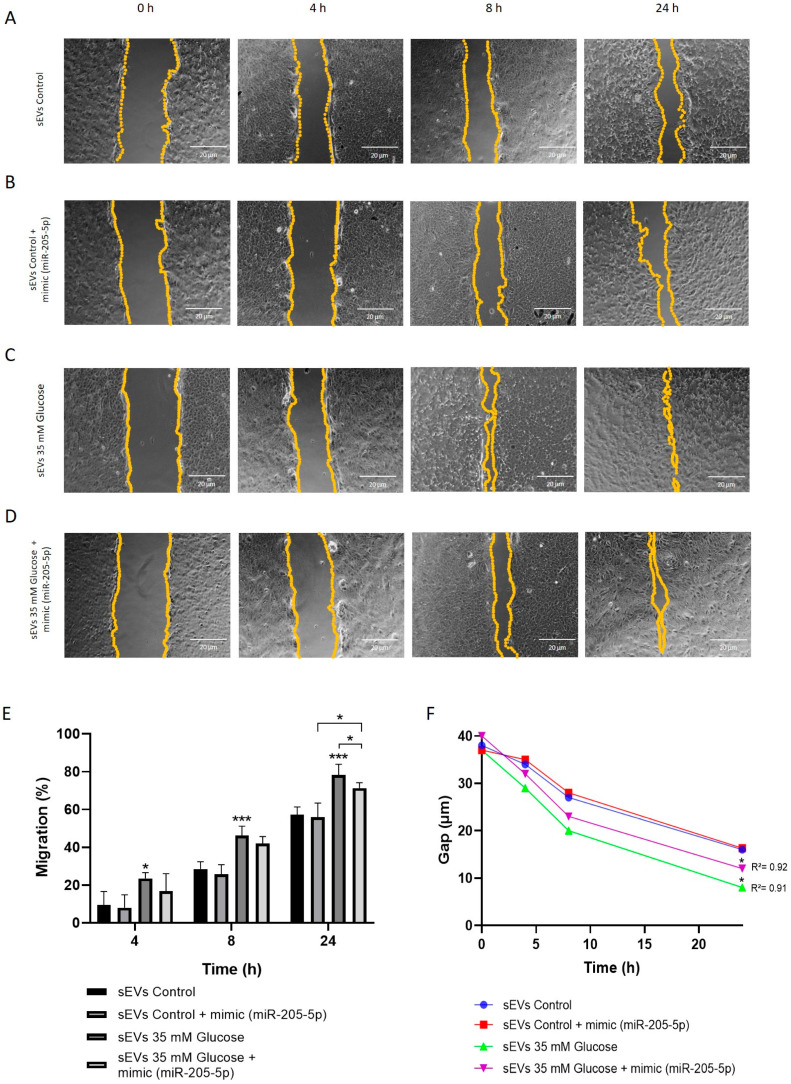
Cell migration was negatively regulated by miR-205-5p sEVs in ARPE-19 cells. Representative images at 0, 4, 8 and 24 h under different conditions: control sEVs (**A**), control + mimic miR-205-5p sEVs (**B**), glucose 35 mM sEVs (**C**) and glucose 35 mM + mimic miR-205-5p sEVs (**D**). Migrated area (%) (**E**) and gap distance (µm) (**F**) were analyzed with Image J (version 1.8.0). Values are expressed as mean ± SEM (*n* = 3). *p*-value was obtained by one-way ANOVA followed by Tukey’s test: * *p* < 0.05 and *** *p* < 0.001 (**E**). The Spearman correlation shows the relationship between time and gap distance (µm) (**F**).

**Figure 6 ijms-25-00934-f006:**
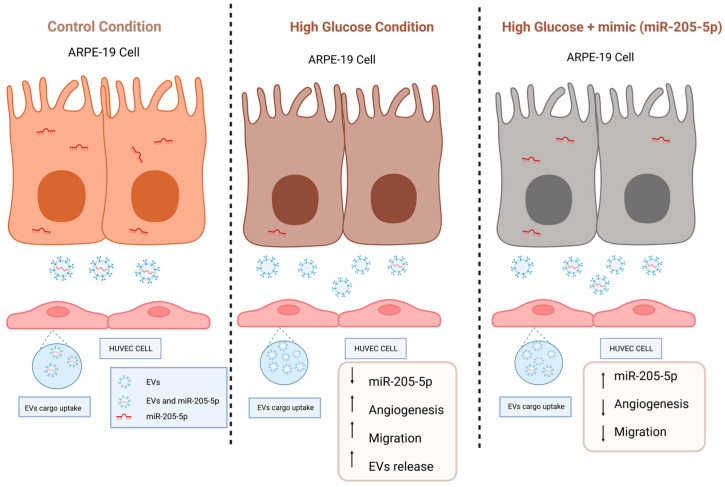
Proposed model illustrating the regulation of exosomal miR-205-5p in angiogenesis, migration and sEVs inclusion under high glucose conditions. The image was created by BioRender.com.

## Data Availability

Data can be provided upon appropriate request.

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
