# Peer review of "Role of Exosomal miR-205-5p Cargo in Angiogenesis and Cell Migration"

_ijms, 2024, doi:10.3390/ijms25020934_

Round 1
Reviewer 1 Report
Comments and Suggestions for Authors
Comments for ijms-2813862
General comments
In this study, the authors investigated the role of small extracellular vesicles (sEVs) containing microRNA-205-5p (miR-205-5p) in intercellular communication under high glucose (HG) conditions. The research focuses on ARPE-19 cells cultured in high glucose for five days, isolating and characterizing sEVs from these cells. The study reveals that HG increases sEVs release but leads to the downregulation of miR-205-5p expression in the cargo compared to the control group. Functionally, sEVs from HG-treated cells promote angiogenesis and cell migration, while miR-205-5p overexpression inhibits these processes. The results demonstrate that ARPE-19 cells respond to HG by increasing sEVs with reduced miR-205-5p cargo, which alone is sufficient to enhance angiogenesis. Conversely, restoring sEVs-miR-205-5p levels mitigates angiogenesis. These findings suggest potential therapeutic applications of sEVs containing miR-205-5p to counteract angiogenesis-related processes under high glucose conditions.
1. What type of t-tests were used for Figures 2 and 3?
2. Figure 5: There is no significant difference in cell migration between the control sEVs and control + mimic miR-205-5p sEVs. This result does not support the conclusion that miR-205-5p inhibits angiogenesis.
Reviewer 2 Report
Comments and Suggestions for Authors
1、 There is a word in the abstract described as microARNs, the body is microRNAs, whether it is a spelling error, please correct. In the section of Materials and Methods, the spelling of CO2 in the last sentence is wrong, and it is suggested that the author carefully correct the whole article to avoid low-level mistakes. What is the meaning of Figure 2BA and 2EB expressed in Chapter 2.2 of this article? Is it spelled wrong or expressed in a wrong way.
2、 In the abstract of this paper, it is proposed that miR-205-5p can promote angiogenesis, but the experiment only in Figure 4 cannot fully prove the theory proposed by the author. It is suggested to increase animal experiments to fully prove the theory proposed in this paper.
3、 For WB experiment in Figure 1D, it is suggested to provide more beautiful experimental results. Among them, whether β-actin is used as an internal reference, why there is a more obvious difference compared with the experimental group.
4、 The paper proposed that miR-205-5p could reduce cell migration, but in Figure 5A and 5B, Figure 5B increased the content of miR-205-5p, but its migration degree was not much different from Figure 5A. Therefore, whether the influence of iR-205-5p on cell migration was as strong as proposed in the paper.
5、 As for Figure 2A and 2B, the nuclear staining brightness of DAPI group and Merge group is significantly different, so it is recommended to unify the brightness to reduce experimental errors.
6、 The authors propose that high glucose can promote the secretion of sev and reduce the load of miR-205-5p. What is the mechanism?
7、 The author's description of the results in Figure 4 is too vague, and the cells in the experimental picture are too small, which makes it impossible to distinguish the results described by the author intuitively. It is suggested to provide clearer experimental results.

The English description of this article needs to be improved, and there are many spelling mistakes, I suggest the author to check carefully
Reviewer 3 Report
Comments and Suggestions for Authors
In their manuscript, Martínez-Santos, Ybarra, et al., investigate the role of exosomal miR-205-5p cargo in modulating angiogenesis and cell migration under high glucose (HG) conditions. They report that HG conditions lead to an increased release of small extracellular vesicles (sEVs), yet paradoxically result in the downregulation of miR-205-5p cargo expression compared to controls. Furthermore, the authors elucidate the impact of these sEVs on ARPE-19 cells, revealing that a reduced level of this microRNA may indeed promote angiogenesis. I propose that the manuscript be accepted contingent upon the implementation of the following amendments:
1. In Figure 1D, the bands representing CD63 lack clarity. The authors should clarify the underlying cause and endeavor to present a clearer depiction of the CD63 bands. Additionally, there is a significant discrepancy between the control sEVs and the sEVs treated with 35mM glucose for β-actin. This observation merits explicit mention and discussion.
2. Figure 2F features a 4-hour mark on the x-axis, the origin of which is unclear. The authors should clarify this temporal designation.
3. Figure 4 assesses tube formation using total length as a metric. However, a comprehensive analysis of angiogenic potential also necessitates consideration of additional parameters such as the total field area, the number of tubules, and the number of junctions. The authors should include these metrics in their analysis.
4. In Figure 6, 'HUVEL' cells should be corrected to 'HUVEC'.
5. Across several cell images, there is a consistent omission of the scale indication. It is critical that the authors address this by adding the appropriate scale to each image.
6. The lack of differential effects on tube formation and cell migration after overexpressing miR-205-5p, even with a 2000-fold increase, is a significant observation. The authors are encouraged to discuss this finding in depth and postulate the underlying mechanisms.
Comments on the Quality of English LanguageThe manuscript contains some spelling errors. Thorough proofreading is imperative to correct all such errors, thus enhancing the manuscript's clarity and professionalism.
Round 2
Reviewer 1 Report
Comments and Suggestions for Authors
The authors have addressed all my concerns. Thank you!